# Factors Influencing Retention among Regional, Rural and Remote Undergraduate Nursing Students in Australia: A Systematic Review of Current Research Evidence

**DOI:** 10.3390/ijerph20053983

**Published:** 2023-02-23

**Authors:** Xian-Liang Liu, Tao Wang, Daniel Bressington, Bróna Nic Giolla Easpaig, Lolita Wikander, Jing-Yu (Benjamin) Tan

**Affiliations:** Faculty of Health, Charles Darwin University, Darwin, NT 0810, Australia

**Keywords:** influencing factors, retention, remote, undergraduate nursing students, Australia

## Abstract

Background: This systematic review aimed to explore the factors influencing retention among regional, rural, and remote undergraduate nursing students who were enrolled in Australian universities. Methods: Mixed-methods systematic review. A+ Education, CINAHL, Education Resources Information Center (ERIC), Education Research Complete, JBI EBP database, Journals@Ovid, Medline, PsycINFO, PubMed, and Web of Science were systematically searched from September 2017 to September 2022 to identify eligible English-language studies. The methodological quality of the included studies was critically assessed using the Joanna Briggs Institute’s critical appraisal tools. Descriptive analysis with a convergent segregated approach was conducted to synthesize and integrate the results from the included studies. Results: Two quantitative and four qualitative studies were included in this systematic review. Both the quantitative and qualitative findings demonstrated that additional academic and personal support was essential for improving retention among undergraduate nursing students from regional, rural, and remote areas in Australia. The qualitative synthesis also highlighted many internal (e.g., personal qualities, stress, ability to engage with classes and institutions, time management, lack of confidence, cultural well-being, and Indigenous identity) and external factors (e.g., technical difficulties, casual tutors, different competing demands, study facilities, and financial and logistical barriers) that influenced retention among undergraduate nursing students from regional, rural, and remote areas in Australia. Conclusions: This systematic review demonstrates that identifying potentially modifiable factors could be the focus of retention support programs for undergraduate nursing students. The findings of this systematic review provide a direction for the development of retention support strategies and programs for undergraduate nursing students from regional, rural and remote areas in Australia.

## 1. Introduction

The problem of a shortage in the nursing workforce is an important challenge for healthcare systems around the world. Due to the COVID-19 pandemic and other factors, such as the ageing nursing workforce, the shortage is likely to be exacerbated [1]. In Australia, registered nurses account for 68% of healthcare professionals in remote and very remote regions [2]. Therefore, the successful completion of nursing degrees is essential for developing an effective healthcare workforce capable of meeting the healthcare needs of Australians who are living in regional, rural, and remote areas. However, not all nursing students choose to complete their studies, and leave university prior to completion. Poor student retention is a worldwide and long-standing issue in nursing education, with significant impacts on the future of the nursing and healthcare workforce [3,4,5]. Between 10% and 40% of nursing students leave their nursing programs in Australia [6,7], and the average retention rate for nursing students is 50% in the United States [8] and about 75% in the United Kingdom [9]. With the growth of online programs after the COVID-19 pandemic, more attributes contribute to poor student retention, such as low engagement in online learning [10]. The issue of poor student retention has raised concerns for students enrolled in nursing programs and for nurse educators [11].

The retention of nursing students is complex, and both internal and external factors in nursing education and in students’ personal lives affect student retention, such as family support, competing obligations, access to university services, and financial issues [12,13,14]. Furthermore, the social context may also affect nursing students’ motivation and influence their decision to stay in or leave their nursing programs [15,16]. The most significant factors in nursing students’ success and retention are provision of tailored services and the delivery of strategies to meet their social needs [17,18,19]. However, nursing students are more likely to focus on personal reasons rather than social factors, which may influence their ability to remain in a program [20]. In Australia, the majority of students who enrolled in the Bachelor of Nursing Program resided in non-metropolitan (regional, rural, and remote) areas [21]. Nursing students from regional, rural, and remote areas face additional challenges and barriers to completing their university nursing studies in a country as vast as Australia [22,23]. These nursing students are often at high risk of dropping out due to additional challenges, including loneliness and isolation if they are away from family support, as well as difficulties with course engagement [24,25,26]. Early identification of these challenges and barriers is essential for promoting nursing students’ success and increasing their retention rates.

Many nursing students leave their studies during the first semester, and the first six weeks have been reported as the essential time period within which many students will withdraw [27]. Understanding why nursing students leave their programs and, connectedly, how to increase student retention is a persistent challenge [28]. Student retention is a critical area of nursing education, and non-metropolitan undergraduate nursing students may face additional retention challenges; however, no systematic review has been conducted to summarize the factors influencing retention among regional, rural, and remote undergraduate nursing students in Australia. One systematic review [5] identified a range of factors influencing student retention in all tertiary teaching courses, but the specific factors relevant to the regional, rural, and remote undergraduate nursing students remains unclear. Another systematic review [14] explored the factors influencing the retention of Aboriginal and Torres Strait Islander students across all tertiary health disciplines, but non-Indigenous nursing students from non-metropolitan areas in Australia were excluded. This is an important gap in evidence synthesis, as a systematic review of factors influencing the retention of non-metropolitan undergraduate nursing students would highlight research gaps and inform the designs of tailored support strategies to help them successfully complete their studies in Australia. Therefore, this systematic review aimed to explore retention issues, particularly the factors influencing retention among regional, rural, and remote undergraduate nursing students who were enrolled in Australian universities.

## 2. Materials and Methods

The systematic review protocol was prospectively registered in the International Platform of Registered Systematic Review and Meta-analysis Protocols (INPLASY) prior to its commencement (registration number: INPLASY202260087). The methods and procedures used in this mixed-methods systematic review were in accordance with the Preferred Reporting Items for Systematic Reviews and Meta-Analyses (PRISMA) reporting guideline [29] and the Joanna Briggs Institute’s (JBI) approach for conducting mixed-methods systematic reviews [30].

### 2.1. Search Methods for Identification of Studies

A+ Education, CINAHL, Education Resources Information Center (ERIC), Education Research Complete, JBI EBP database, Journals@Ovid, Medline, PsycINFO, PubMed, and Web of Science were systematically searched from September 2017 to September 2022 to locate eligible English-language publications. The following Medical Subject Headings terms with related keywords were used to develop comprehensive search strategies for these 10 databases to identify eligible studies (see Appendix A): “student dropouts”, “attrition”, “retention”, “academic performance”, “undergraduate student*”, “baccalaureate nurs*”, “regional”, “remote”, “rural”, “influencing factor*”, “barrier*”, “facilitator*”, “challenge*”, “difficult*”, and “Australia*”. Reference lists of the included studies were reviewed, and Google Scholar was used to identify publications that had cited the included studies.

### 2.2. Eligibility Criteria

The inclusion criteria applied to study selection included:
(1)Population: Undergraduate nursing students from non-metropolitan (regional, rural, and remote) areas of Australia, defined by Australian Standard Geographical Classification (ASGC), were recruited as part or all of the participants in the studies. The non-metropolitan (regional, rural, and remote) areas were all areas outside Australia’s major cities [31].(2)Phenomena of interest: Studies containing qualitative or quantitative findings pertaining to the factors influencing retention among non-metropolitan (regional, rural, and remote) undergraduate nursing students. Qualitative, quantitative, or mixed-method studies were included if their aims or objectives were to explore the experiences, perceptions, and perspectives regarding the factors influencing retention. Student retention referred to students who enrolled in university programs, successfully passed the academic period, and avoided dropping out, interruption, or withdrawal [32].(3)Context: The higher education sector in Australia.(4)Types of studies: Original qualitative, quantitative, and mixed-methods studies published or available in English in the last five years (September 2017 to September 2022) to provide a current overview of the literature. Descriptive or discussion articles, review articles, editorials, letters, commentaries, and conference abstracts were excluded.

### 2.3. Study Selection

All records from the databases were imported into EndNote (version X9). Two-stage data screening was undertaken: initial database searches were performed by one reviewer (XL-L), and duplicate records were removed by EndNote before two reviewers (XL-L and WT) reviewed the titles and abstracts independently. Subsequently, full-text assessments were conducted by two reviewers (XL-L and WT) independently. Full-text publications that did not meet the eligibility criteria were excluded. Any disagreements during the data screening process were resolved by discussion with the review team members.

### 2.4. Data Extraction

Data extraction forms from the JBI were used to extract data from the included studies. The following data were extracted from the included studies: author(s), publication year, journal, study question/aim, population, geographical location of the study, sample size, data collection, analysis, and all findings or results related to the factors influencing retention. One reviewer (XL-L) performed data extraction, which was double-checked by a second reviewer (WT).

### 2.5. Quality Appraisal

Methodological quality was critically assessed for each of the included studies using the JBI critical appraisal tools (https://jbi.global/critical-appraisal-tools, accessed on 15 June 2022), including the cross-sectional studies critical appraisal checklist and qualitative studies critical appraisal checklist. The JBI has different critical appraisal tools for different study designs. One reviewer read the included publications and answered all the questions on the quality appraisal checklists. A second reviewer checked all the quality appraisal results. Discrepancies were discussed with the review team if required until consensus was achieved. 

### 2.6. Data Synthesis

Descriptive analysis with a convergent segregated approach was used for the synthesis of the results from the included studies; separate quantitative and qualitative data syntheses were performed, followed by the integration of the findings derived from the quantitative and qualitative syntheses [30]. The factors influencing retention were identified in each of the included studies. The quantitative results were translated into a narrative description, thus allowing integration with the qualitative findings from the included studies. This review obtained a greater depth of understanding of the factors influencing retention among regional, rural, and remote undergraduate nursing students in Australia by integrating the quantitative and qualitative synthesized results and linking the two sets of data. The data syntheses were performed by one reviewer (XL-L), with regular checking and input from the review team.

## 3. Results

Six studies were included in this systematic review, and the reasons for exclusion are reported in Figure 1. Moreover, this systematic review also incorporated a search of the reference lists of the six included studies. Three potential records from the reference lists were assessed, but no study was included. As a result, a total of six records [26,33,34,35,36,37] were included in this systematic review. The overall process of the literature screening is illustrated by the PRISMA flow diagram shown in Figure 1.

### 3.1. Characteristics of Included Studies

The final synthesis was constructed from the two quantitative studies [34,35] and four qualitative studies [26,33,36,37] (see Table 1). Five reported primary research findings in peer-reviewed journals [26,33,34,35,36], and one was a Master of Science (Research) thesis [37]. The studies were conducted in Queensland (n = 3) and New South Wales (n = 3), Australia. Convenience or purposive sampling was notably favored in all the included studies, with sample sizes ranging from 4 [37] to 4472 [35]. The 6 included studies recruited 4816 undergraduate nursing students: the 2 quantitative studies [34,35] combined recruited 4673 participants, and 3 of the qualitative studies [26,36,37] had less than 10 participants each. One study [37] recruited Indigenous nursing students only. The instruments used in the two quantitative studies [34,35] included a questionnaire developed by the authors and administrative data from the Student Management System, respectively. A descriptive qualitative design [26,33,36] and a phenomenological study design [37] were used in the included qualitative studies. Moreover, three of the qualitative studies [26,36,37] used interviews and one study [33] used an open-ended electronic questionnaire. 

The two quantitative studies [34,35] explored the factors influencing students’ retention following the implementation of a retention intervention and a support program, respectively. One quantitative study [35] provided the Professional Communication Academic Literacy (PCAL) support program for undergraduate nursing students and explored its impact on the students’ engagement. The PCAL support program was used to improve the capacities of all university students who sought discipline-relevant academic writing skills support [35]. The other quantitative study [34] aimed to explore undergraduate nursing students’ intention to complete their nursing program after implementing the Initiatives for Retention (IR) intervention. The IR intervention was an innovative strategy, and it was designed based on the four pillars of student engagement: behavioural, affective, social, and cognitive engagement [38].

### 3.2. Quality Appraisal

The JBI’s analytical cross-sectional studies critical appraisal tool was used for the two quantitative studies [34,35], and the JBI’s qualitative research critical appraisal tool was used for the four qualitative studies [26,33,36,37], as shown in Table 2. Half of the JBI’s qualitative critical appraisal checklist items were “unclear” in one qualitative study [33]. Most of the critical appraisal checklist items were confirmed as “Yes” in the other three qualitative studies [26,36,37], but the relationship between the nursing students and the data collectors was commonly not described. Neither of the quantitative studies [34,35] used objective or standard criteria as a condition measurement as all the participants were nursing students. In addition, the strategies used to deal with the effects of confounding factors in the study design and data analysis of the two quantitative studies [34,35] were “unclear”. No studies were excluded based on the quality appraisal, which was conducted for informational purposes only.

### 3.3. Factors Influencing Retention

The results of the two quantitative studies [34,35] included the factors influencing the retention of undergraduate nursing students following the implementation of the IR intervention and the PCAL support program, respectively. There was 1 quantitative study [35] that had 4472 participants and reported that the undergraduate nursing students who had accessed PCAL academic support were more likely to continue to remain enrolled in their nursing program (95.2%, n = 2092) compared with non-PCAL consult students (72.4%, n = 3090), *p* < 0.001. On the other hand, simultaneous logistic regression analysis results also supported that students who continued to enroll in the nursing program were more likely to seek PCAL support (adjusted odds ratio: 7.81, 95% CI: 6.18 to 9.86, *p* < 0.001). Moreover, that study [35] also identified seven variables as predictors of high academic performance in undergraduate nursing students who had remained enrolled in their program for at least one year, such as age (21 years or older), gender (female), Australian-born, domestic students, sought PCAL academic support, lived in high socioeconomic-status residential areas, and were not the first in their family to attend university. In the other quantitative study [34], a prospective cross-sectional survey was conducted among 201 participants to evaluate the perception of belonging to the School of Nursing and the nursing students’ intention to complete their nursing program after participating in the IR intervention. That study [34] reported that 30% of the students had considered leaving their nursing program in the previous year, but, after receiving extra personal support, they continued with the program. 

This systematic review included four qualitative studies [26,33,36,37]. One of the three studies [26] that used interviews explored nursing students’ experiences at a remote Australian university campus and the barriers that greatly impacted their ability and motivation to study. Many factors impacted the nursing students’ learning experiences and retention, including stress experienced, the ability to engage with classes, technical difficulties in videoconference lectures, recorded lectures rather than a live videoconference, casual tutors prioritizing their full-time jobs, and small-cohort courses [26]. Nursing students were positive about face-to-face teaching and small cohort size, which enabled the students to create supportive and strong relationships with other students and teaching and support staff [26]. The participants also described many other barriers, including the lack of support from partners, family, friends, and the community, difficulties in developing mentor–mentee relationships, and the lack of access to study facilities outside of business hours, significantly influenced their motivation to study [26]. Furthermore, the participants described financial and logistical issues faced in completing mandatory placements and travelling between university campuses [26].

Another qualitative study [36] that used interviews explored the experiences of first-year nursing students who had failed one or more units but continued with their nursing courses. The participants shared their struggles and the factors influencing their aim to meet academic requirements and pass their repeating nursing unit(s), including academic support and resources, different competing demands that affected their studies, lack of confidence in academic writing, lack of motivational support from family members and peers, working hours, and family duties, as well as fortitude and resilience (i.e., to accomplish their dream to be a nurse) [36].

The remaining qualitative study [37] that used interviews explored the experiences of a group of Indigenous nursing students and the insight strategies used to improve retention and completion rates. The themes identified in this study were misperceptions, cultural well-being and Indigenous identity, and internal conflict [37]. The most prominent misperceptions were that Indigenous students learned differently and that Indigenous nursing students had similar barriers to their studies compared with non-Indigenous students, including family demands and commitments, financial restraints, knowledge limitations, and lack of study skills and time management [37]. However, the Indigenous nursing students also reported that they had experienced “feeling different” (p. 37) as a barrier to completing their studies, which challenged their ideations as Indigenous nursing students, and “feeling different” (p. 37) included cultural well-being and Indigenous identity [37]. The participants in that study [37] also described “internal conflict” (p. 41), which included a lack of engagement with other students and the institution, as well as the community, and cultural responsibility and conflict. The Indigenous nursing students also struggled with managing commitments to their families, community, culture, and university [37]. 

Work integrated learning (WIL) or placement is an integral part of the studying journey for bachelor’s degree nursing students in Australia. The qualitative study [33] that used an open-ended electronic questionnaire identified six personal qualities that supported undergraduate nursing students in achieving WIL or placement success, including confidence, commitment to learning, compassion, effective communication skills, enthusiasm to succeed, and self-motivation.

The independent syntheses of the quantitative and qualitative findings largely supported each other. The narrative description of the quantitative findings indicated that extra support was essential for improving retention among regional, rural, and remote undergraduate nursing students in Australia. This was supported by the qualitative findings in which most of the qualitative studies uncovered the importance of support in improving undergraduate nursing students’ retention and learning experiences. For example, one qualitative study [26] found that different kinds of support were extremely important for nursing students at remote Australian university campuses, including available support services and facilities from the university, as well as support from partners, family, friends, and the wider community. However, the factors influencing retention among regional, rural, and remote undergraduate nursing students was not the main purpose of the quantitative studies [34,35], and very limited data related to the factors influencing retention were identified.

## 4. Discussion

Understanding the factors that impact nursing students’ retention is essential for improving retention rates in nursing programs [14,39]. This systematic review is the first to synthesize findings from quantitative and qualitative studies to further our knowledge of the factors influencing retention among regional, rural, and remote undergraduate nursing students in Australia. Despite the pressing need for more nurses in Australia, few studies have explored the issues and factors that explicitly impact retention among undergraduate nursing students from regional, rural, and remote backgrounds in Australia.

### 4.1. Key Findings and Implications

Both the quantitative and qualitative findings demonstrated that extra academic and personal support was essential for improving retention among undergraduate nursing students from regional, rural, and remote areas in Australia [26,33,34,35,36,37]. Moreover, the qualitative synthesis also highlighted many internal (e.g., personal qualities, stress, ability to engage with classes and institutions, time management, lack of confidence, cultural well-being, and Indigenous identity) and external factors (e.g., technical difficulties, casual tutors, different competing demands, study facilities, and financial and logistical barriers) that influenced retention among undergraduate nursing students from regional, rural, and remote areas in Australia [26,33,36,37]. These factors are useful for informing the efforts and support needed to improve the retention rates of undergraduate nursing students from regional, rural, and remote backgrounds in Australia. Efforts and support strategies should begin before students start their nursing program, or at least within the first six weeks, as many students may withdraw during this essential time period [27]. The findings of this systematic review were consistent with those in previous national and international reviews [13,14,20]. Despite these findings, there were few support strategies used by nursing educators to improve the retention of undergraduate nursing students as shown by the minimal documented evaluation of support programs and limited evidence about which support programs were most effective [14,20]. 

This systematic review included two quantitative studies [34,35] that found that an IR intervention and the PCAL support program implemented by schools and faculties appeared to enhance the retention of undergraduate nursing students. Other support programs implemented by universities have also improved nursing student retention as reported by a recently published review [39], including the Academic, Personal and Professional Learning (APPL) support program, an automated text messaging system, an extracurricular student support group, and a pastoral care support adviser service. These support programs and services assisted the nursing students in improving their self-confidence and motivation and encouraged them to believe in their ability to complete their courses [39]. However, that review [39] only included five mixed-methods studies, four qualitative studies, and one case study, and no intervention trials were included. Most of the included studies in that recent review [39] were from the United Kingdom, and no studies were conducted in Australia. 

In addition, support from family and peers was the most frequently reported support for nursing student retention, as mentioned by three qualitative studies [26,36,37] in the present review. Family and peer support (e.g., emotional, financial, and physical support) gives nursing students confidence and encourages them to remain in their study programs [14]. Moreover, nursing students’ support-seeking behaviors have been associated with both progression and retention, and nursing students who seek academic and social support are less likely to withdraw [15].

University schools and faculties have an essential role to play in addressing dropout and attrition issues among undergraduate nursing students by using different retention strategies. This systematic review identified many internal and external factors that influenced retention among undergraduate nursing students from regional, rural, and remote areas in Australia. Similarly, one study [25] developed a matrix layout to describe the barriers to and enablers of retention among Aboriginal nursing students, and the main themes were resilience (e.g., support networks), organizational support (e.g., financial support and family support), vulnerability (e.g., lack of confidence), and shame (e.g., family priorities and academic preparation). These factors were very important in informing the development of retention strategies and programs for undergraduate nursing students from regional, rural, and remote areas. A previous scoping review summarized all the current retention strategies that have been used in nursing programs worldwide, and most of the components were mentorship, literacy and language approaches, study skills, and tutoring [28]. That review [28] also found that “whole-program strategies” with pathway options initiated at the recruitment or pre-program phase were the most effective retention strategies for nursing students. 

Undergraduate nursing students from remote or Indigenous backgrounds faced some unique barriers to completing their nursing programs compared with urban nursing students, such as racism, discrimination, and disconnection [14,37]. Furthermore, for many of the students from remote or Indigenous backgrounds studying nursing at satellite university campuses in Australia, poor support and a lack of educational resources and services led to increased withdrawal from their courses [40]. To address the unique barriers and perspectives of students from remote or Indigenous backgrounds, previous systematic reviews [14,40] identified some successful practical strategies to improve the retention of Indigenous healthcare students. The suggested strategies were multilayered and were initiated before the students commenced at their universities, such as culturally appropriate recruitment, selection procedures, and mentoring services; appointing Indigenous lecturers; embedding Indigenous content within the curriculum; developing tutoring and mentoring programs; and working with the Indigenous Student Support Centre [14,40]. Other original studies [25,41] have suggested that organizational support along with problem solving, flexibility, and peer cohesion, as well as mentoring circles (including the university environment, group identity, and emotional intelligence) could improve Aboriginal nursing students’ retention in their courses.

### 4.2. Limitations

This systematic review has some limitations. Firstly, this systematic review only included original studies that were published in the last five years and conducted in Australia, and thus some studies published before September 2017 and conducted in other countries with non-metropolitan nursing students may have been overlooked. Secondly, the included studies were variable in study objectives, study population, and methodological quality. For example, this systematic review included studies that recruited undergraduate nursing students from regional, rural, and remote areas as part or all of the participants in the studies, and thus some studies may also have included urban nursing students. In addition, exploring the factors influencing retention was not the main purpose of the two included quantitative studies, and they included very limited findings related to the factors influencing retention [34,35]. 

## 5. Conclusions

Many factors influenced the retention of regional, rural, and remote undergraduate nursing students in Australia, and the key influencing factors included extra support (e.g., academic support, family support, and peer support), personal qualities (e.g., confidence, self-motivation, and resilience), engaging with classes and institutions, different competing demands, financial barriers, cultural well-being, and Indigenous identity. The findings of this systematic review provide a direction for the development of retention and support strategies and programs for undergraduate nursing students from regional, rural, and remote areas in Australia.

## Figures and Tables

**Figure 1 ijerph-20-03983-f001:**
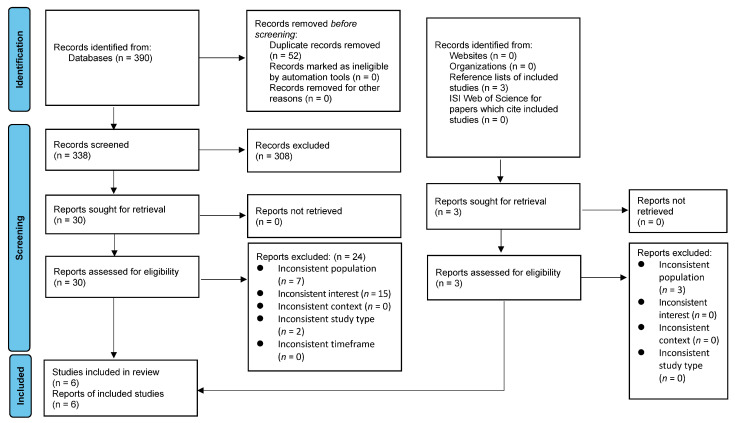
PRISMA flow diagram for search results.

**Table 1 ijerph-20-03983-t001:** Characteristics of the included studies.

First Author, Year	Context and Settings	Study Design and Data Source	Participants (n)	Study Objective Related to This Systematic Review	Influencing Factors and Barriers to Retention	Retention Rate/Intention
Hays, C., 2022 [26]	A regional university in Queensland, Australia	Qualitative descriptive study, interviews	Undergraduate nursing students (n = 9)	Explore the learning experiences of studying at a university campus located in a remote town	Physical stress felt because of other competing priorities Ability to engage with study Experiences of face-to-face tutorials and recorded or videoconference lectures (technical difficulties and casually employed staff prioritizing their full-time positions at the hospital) Forming mentor–mentee relationships difficulties Lack of support from partners, family, friends, and the wider community Lack of financial support	N/A
Walker, S. B., 2021 [33]	A regional university in Queensland Australia	A descriptive exploratory study design, electronic questionnaire	Undergraduate nursing students (n = 121)	Identify the personal qualities the students bring to and need for work integrated learning success	Six personal qualities: confidence, commitment to learning, compassion, effective communication skills, enthusiasm to succeed, and self-motivation.	N/A
Middleton, R., 2021 [34]	A regional university with five satellite campuses and a main campus across New South Wales, Australia	A prospective cross-sectional study, university belonging questionnaire	Pre-registration nursing students enrolled in the undergraduate nursing program (n = 201)	Evaluate students’ intention to complete the pre-registration undergraduate nursing program following the initiatives for retention intervention	Extra personal support	84% of the participants plan to continue and finish their study.
Glew, P. J., 2019 [35]	A large multi-campus program in Australia	A correlational design, administrative data	First-, second- and third-year undergraduate nursing students (n = 4472)	Examine support uptake and the demographic characteristics of nursing students who sought support, and the relationships between demographic characteristics with undergraduate student retention	Students who sought Professional Communication Academic Literacy (PCAL) support were more likely to continue to remain enrolled (72.4% versus 95.2%, *p* < 0.001). Predictors of continuing, high academic performance nursing students: non-school leavers, female, Australian-born, domestic students, not first-in-family to attend university, classified to have residential addresses in high socio-economic status (SES) areas and those who sought Professional Communication Academic Literacy (PCAL) support.	72.4 % for non-support consult students and 95.2% for support consult students
Elmir, R., 2019 [36]	A large nursing school in New South Wales, Australia	A descriptive qualitative design, interviews	First year undergraduate nursing students who repeating one or more units (n = 9)	Explore nursing students’ experiences of repeating at least one unit	Lack of academic writing confidence Many competing demands that affected their studies Working hours and family duties Academic support and resources (peer support) Motivational support from family members Fortitude and resilience (accomplish dream)	N/A
Henschke, K., 2017 [37]	A regional university at south-east Queensland	A phenomenological study, in-depth interviews	Indigenous Bachelor of Nursing students (n = 4)	Explore insight strategies to improve retention, attrition and completion rates.	Misperceptions: Preparedness (knowledge limitations) and similarities (family demands and commitments, financial restraints, knowledge limitations, lack of study skills and time management) Feeling different: Cultural well-being and indigenous identity Internal conflict: Lack of engagement, cultural or community responsibility and conflict	N/A

Note: N/A: not applicable.

**Table 2 ijerph-20-03983-t002:** Methodological quality assessment of included studies.

First Author, Year	Item 1	Item 2	Item 3	Item 4	Item 5	Item 6	Item 7	Item 8	Item 9	Item 10
Hays, C., 2022 [26]	Yes	Yes	Yes	Yes	Yes	Yes	Unclear	Yes	Yes	Yes
Walker, S. B., 2021 [33]	Unclear	Unclear	Unclear	Yes	Yes	Unclear	Unclear	Yes	Yes	Yes
Middleton, R., 2021 [34]	Yes	Yes	NA	No	Yes	Unclear	Yes	Yes	---	---
Glew, P. J., 2019 [35]	Yes	Yes	NA	No	Yes	Unclear	No	Yes	---	---
Elmir, R., 2019 [36]	Yes	Yes	Yes	Yes	Yes	Unclear	Unclear	Unclear	Yes	Yes
Henschke, K., 2017 [37]	Yes	Yes	Yes	Yes	Yes	Unclear	Unclear	Unclear	Yes	Yes

Note: NA: Not applicable; the JBI analytical cross-sectional studies critical appraisal tool (Item 1 to 8: “1. Were the criteria for inclusion in the sample clearly defined?” “2. Were the study subjects and the setting described in detail?” “3. Was the exposure measured in a valid and reliable way?” “4. Were objective, standard criteria used for measurement of the condition?” “5. Were confounding factors identified?” “6. Were strategies to deal with confounding factors stated?” “7. Were the outcomes measured in a valid and reliable way?” “8. Was appropriate statistical analysis used?”) was used for the quantitative studies and the JBI qualitative research critical appraisal tool (Item 1 to 10: “1. Is there congruity between the stated philosophical perspective and the research methodology?” “2. Is there congruity between the research methodology and the research question or objectives?” “3. Is there congruity between the research methodology and the methods used to collect data?” “4. Is there congruity between the research methodology and the representation and analysis of data?” “5. Is there congruity between the research methodology and the interpretation of results?” “6. Is there a statement locating the researcher culturally or theoretically?” “7. Is the influence of the researcher on the research, and vice- versa, addressed?” “8. Are participants, and their voices, adequately represented?” “9. Is the research ethical according to current criteria or, for recent studies, and is there evidence of ethical approval by an appropriate body?” “10. Do the conclusions drawn in the research report flow from the analysis, or interpretation, of the data?”) was used for qualitative studies (https://jbi.global/critical-appraisal-tools, accessed on 15 June 2022).

## Data Availability

Not applicable.

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
