# Peer review of "Factors Influencing Retention among Regional, Rural and Remote Undergraduate Nursing Students in Australia: A Systematic Review of Current Research Evidence"

_ijerph, 2023, doi:10.3390/ijerph20053983_

Round 1

Reviewer 1 Report

Thank you for giving me the opportunity to review this manuscript entitled: " Factors influencing retention among regional, rural and remote undergraduate nursing students in Australia: A systematic review of current research evidence".

This manuscript presents a novel study. And I believe the need for it is well justified and points to the difficulties present in the retention of undergraduate nursing students in Australia.

The data from this study, although very specific, is interesting in order to implement actions to retain students, taking into account the different barriers that they may present. However, I believe that certain aspects of your manuscript could be improved. Therefore, below you will find my comments and suggestions which I believe can improve the final quality of the document.

- Abstract: I would remove the total n from all studies, I think this is more appropriate for a meta-analysis.  

- The introduction seems adequate and is very clear. It has set out the current situation where there are problems in the retention of students in different areas, and this context has been addressed in a general way and in particular by pointing out the situation in Australia.

-Methods:

- Figure 1 shows the results of your search, this section should be considered within the Results section and not included in methodology.

- In methodology it is necessary to point out the methods used to assess the risk of bias due to missing results in a synthesis.

- Results: Line 200-204. The information provided here overlaps with the PRISMA figure, I think it would be preferable to refer to the figure and indicate that in the end there have been six studies included for review.

- I do not quite understand what is being referred to in lines 205-208. It is not clear to me whether they have extracted any studies from their references or not.

- In Table 2, when reference is made to quality appraisal, I think they should include either in the legend or in the table itself, what the items in the table mean.

- I do not know what is meant by (p.37) and (p.41) in lines 322 and 324.

- The major concern of this manuscript is the wording of its results. I do not understand the division by quantitative and qualitative results. For example, in line 271, is it necessary to talk about specific statistical data of a study for which we do not have all the information?  

- I think the correct thing to do in this case would be, if appropriate, to make sub-headings relating to characteristics of interest, for example, factors influencing student retention, not quantitative and qualitative studies because, in the end, we are dealing with a narrative systematic review.

- On line 272, what are the seven predictor variables of high academic performance?

- Disussion: 363-375 reference is missed.

Reviewer 2 Report

The paper is strong in methodology and writing. The main purpose of this paper is to provide a systematic review demonstrates that identifying potentially modifiable factors could be the focus of retention support programs for undergraduate nursing students.

My one minor concern is that there are no identified strategies specifically for the first 6 weeks of education. There needs to be a better clearer statement on what change/factors need to be focused on during the beginning of nursing educaiton.

Round 2

Reviewer 1 Report

The authors have provided answers to all the concerns raised, and have implemented changes to the manuscript, improving its final quality. At this point, I believe that all the modifications that needed to be implemented have been made. Therefore, my congratulations to the authors for their work.